# Structural Refinement of 2,4-Thiazolidinedione Derivatives as New Anticancer Agents Able to Modulate the BAG3 Protein

**DOI:** 10.3390/molecules27030665

**Published:** 2022-01-20

**Authors:** Dafne Ruggiero, Stefania Terracciano, Gianluigi Lauro, Michela Pecoraro, Silvia Franceschelli, Giuseppe Bifulco, Ines Bruno

**Affiliations:** Department of Pharmacy, University of Salerno, Via Giovanni Paolo II 132, 84084 Fisciano, Italy; druggiero@unisa.it (D.R.); sterracciano@unisa.it (S.T.); glauro@unisa.it (G.L.); mipecoraro@unisa.it (M.P.); sfranceschelli@unisa.it (S.F.); bifulco@unisa.it (G.B.)

**Keywords:** BAG3 protein, chaperones, cancer, virtual screening, 2,4-thiazolidinedione scaffold, BAG domain modulators, surface plasmon resonance (SPR) assay

## Abstract

The multidomain BAG3 protein is a member of the BAG (Bcl-2-associated athanogene) family of co-chaperones, involved in a wide range of protein–protein interactions crucial for many key cellular pathways, including autophagy, cytoskeletal dynamics, and apoptosis. Basal expression of BAG3 is elevated in several tumor cell lines, where it promotes cell survival signaling and apoptosis resistance through the interaction with many protein partners. In addition, its role as a key player of several hallmarks of cancer, such as metastasis, angiogenesis, autophagy activation, and apoptosis inhibition, has been established. Due to its involvement in malignant transformation, BAG3 has emerged as a potential and effective biological target to control multiple cancer-related signaling pathways. Recently, by using a multidisciplinary approach we reported the first synthetic BAG3 modulator interfering with its BAG domain (BD), based on a 2,4-thiazolidinedione scaffold and endowed with significant anti-proliferative activity. Here, a further in silico-driven selection of a 2,4-thiazolidinedione-based compound was performed. Thanks to a straightforward synthesis, relevant binding affinity for the BAG3BD domain, and attractive biological activities, this novel generation of compounds is of great interest for the development of further BAG3 binders, as well as for the elucidation of the biological roles of this protein in tumors. Specifically, we found compound **6** as a new BAG3 modulator with a relevant antiproliferative effect on two different cancer cell lines (IC_50_: A375 = 19.36 μM; HeLa = 18.67 μM).

## 1. Introduction

The multimodular BAG3 protein (Bcl-2-associated athanogene 3) is a member of the human BAG family of co-chaperones that interacts, through its conserved BAG domain (BD) located at the C-terminus of the protein, with the ATPase (Adenosine Triphosphatase) domain of heat shock protein 70 (Hsp70), modulating a variety of physiological and pathological functions [1,2,3,4]. In addition to BD, BAG3 also includes proline rich (PxxP) and WW modules responsible for its interaction with several signaling factors highly involved in cancer development. Through the so-called M-domain, located between PxxP and WW motifs, BAG3 can also mediate the recruitment of many protein kinases implicated in various signal-transduction pathways. Moreover, the BAG3 role as a key player in cellular macroautophagy, mediated by the two conserved IPV (Ile-Pro-Val) motifs, has been reported [5]. Under physiological conditions, normal cells express only low basal levels of BAG3, while its expression is highly induced under several stressful stimuli, where it is essential for maintaining cellular proteostasis by regulating the two key cellular protein degradation pathways such as proteasomal delivering and autophagy [6]. In addition, BAG3 has recently emerged as an attractive target in several pathological conditions, including cancer, driving the main hallmarks of malignancy, such as apoptosis suppression and oncogenic transformation [7]. Indeed, BAG3 is over-expressed in several neoplastic cell types and solid tumors including breast cancer, human hepatocellular carcinoma (HCC), glioblastomas, and pancreatic adenocarcinomas, where its high expression levels are correlated with a poor prognosis [8,9,10,11,12,13,14,15,16]. Recently, a high level of BAG3 has been detected in the serum of pancreatic ductal adenocarcinoma (PDAC) suffering patients, where it interacts with the transmembrane receptor IFITM-2 (interferon-induced transmembrane protein 2), located on macrophages surface, producing the inhibition of IL-6 cellular release [17]. This finding, thus, suggests its relevant role in human diseases linked to macrophages activation, such as inflammatory, cardiac, immune, and degenerative disorders.

All this evidence highlights the crucial value of BAG3 modulation in the field of tumor pathologies and its potential impact on the development of future anticancer strategies. Despite many efforts that have been invested in the exploration of potential BAG3 modulators, the progress in this research area has been hampered by the current unavailability of crystallized structures of BAG3 in complex with binders. Recently, in our multidisciplinary investigation aimed at disclosing new small molecules able to interfere with BAG3 protein, we succeeded in identifying the first synthetic selective BAG domain modulator **1** (Figure 2) endowed with a relevant affinity and selectivity for BAG3BD. Furthermore, compound **1** showed antiproliferative activity in different cancer cell lines (PC3: IC_50_ = 48.1 ± 1.5 µM and A549: IC_50_ = 32.3 ± 0.9 µM) [18].

## 2. Results and Discussion

### 2.1. In-Silico Screening

Starting from the encouraging results obtained in our previous work [18], and with the aim of identifying novel 2,4-thiazolidinedione-based BAG3BD modulators, a fast virtual screening campaign was performed. Specifically, we re-evaluated the virtual library of 2,4-thiazolidinedione-based compounds, built following the related chemical route and accounting the commercial availability of the starting reagents (for further details, please see the section Materials and Methods and [18]). This starting set of compounds was filtered selecting those featured esters and amide chemical functions at N-3 that were considered essential structural motifs for BAG3BD interaction, as emerged in our previously identified modulators [18]. Afterward, this subset of molecules was further refined selecting those featured promising pharmacokinetic properties (see Materials and Methods). Then, we focused on a restricted collection of 2,4-thiazolidinedione derivatives exploiting the convenient synthetic route, and the commercial availability of cheap synthons.

Eventually, with the aim of selecting further promising BAG3BD modulators, we here performed an in-silico screening following a fast ligand-based approach. Specifically, the focused subset of compounds was screened against the lead compound **1** computing the related shape similarity values (Phase software) (Schrodinger, LLC, New York, NY, USA) [19]. In this way, a restricted set of compounds was selected from the original library with a shape comparable to that of the lead compound **1**, following the principle that similar molecules could likely show similar binding modes on the protein counterpart, as demonstrated in several virtual screening studies (Figure 1) [20,21]. Specifically, this protocol envisaged a first conformational search round on each item of the accounted library of 2,4-thiazolidinedione-based compounds and, afterwards, the obtained conformers were aligned and compared to the reference compound **1**. In detail, the “shape similarity” can range from zero (no one atom matching between screened and reference compounds) to one (all atoms matching), and the screened subset of compounds were here further selected only saving those features a shape similarity higher than 0.500.

Using this further filter, the starting library of ~2.3 × 10^4^ 2,4-thiazolidinedione-based compounds was reduced to a small set of eight promising molecules (**2**–**9**), also chosen in order to define some structural information coupled with the outcomes of subsequent biological investigations (Figure 2).

### 2.2. Synthesis of 2,4-Thiazolidinedione Derivatives **2**–**9**

To obtain the newly selected derivatives (**2**–**9**) of our previously described lead compound **1**, we took advantage of the synthetic strategy already optimized by us, shown in Figure 1. Thus, a Knoevenagel condensation has been carried out, in basic conditions for piperidine, between the commercially available 2,4-thiazolidindione **10** and the selected aromatic aldehydes (**a**–**d**), affording the (*Z*)-5-arylidene-2,4 thiazolidinediones **10a–10d** in high yields (Figure 1) [22].

Treatment of **10a**–**10d** with halides (**e**–**g**), in presence of NaH and DMF (Dimethylformamide) dry, provided the desired compounds **2**–**4** and **6**–**9** along with the intermediate **11** [23]. Finally, for the synthesis of compound **5**, a next step in which **11** was coupled with morpholine, in the presence of HOBt and *N*,*N*-diisopropylcarbodiimide (DIC), as coupling agent, was necessary (Figure 1) [24,25].

### 2.3. Biophysical Assays

The synthesized molecules **2**–**9** have been subjected to biophysical screening to evaluate their binding affinities to BAG3. A surface plasmon resonance (SPR) assay [18], using a recombinant BAG3 protein, was performed and, to explore the potential selectivity of our potential binders, the affinity for BAG4—the most closely BAG3 related protein—was also evaluated. Moreover, the binding affinity towards the BAG3BD domain was also assessed. The lead compound **1** was used in the same experimental condition in SPR assay, as reported in Table 1.

All the tested molecules, except for **3** and **4**, exhibited high-affinity binding for the full-length protein with K_D_ values in the nanomolar range. In particular, the most promising compounds were **5** (K_D_ = 12.4 ± 1.2 nM), **6** (K_D_ = 6.3 ± 0.3 nM), and **9** (K_D_ = 24.7 ± 1.8 nM) and, among these, compound **6** showed the best binding profile comparable with that of the lead compound **1** (K_D_ = 11.1 ± 3.9 nM). In addition, to verify the interaction with the BD functional domain of BAG3, compounds **2**, **3**, **5**–**9** were also tested by SPR assay on isolated BAG3BD. Analyzing the data obtained, **6** (K_D_ = 27.6 ± 1.9 nM), **7** (K_D_ = 45.7 ± 1.7 nM), and **9** (K_D_ = 22.4 ± 0.7 nM) exhibited good binding affinities as reported in Table 1, confirming, thus, our computational outcomes. Interestingly, none of the synthesized molecules **2**–**9** was shown to bind BAG4 protein, thereby displaying a high selectivity for BAG3.

### 2.4. Biological Assays

Encouraged by the binding affinities profile obtained for this second collection of 2,4-thiazolidinedione derivatives as BAG3 modulators, we decided to further explore their potential anticancer activity.

Firstly, the antiproliferative activity of lead compound **1**, **2** and **5**–**9** against selected human cell lines was assessed. No effect on human PHA (Phytohaemagglutinin)-stimulated proliferating non-tumor human cell line (PMBC) has been detected for compound **1** tested at different concentrations (10–50 μM) after 72 h of exposure [18]. In more detail, all the synthesized compounds have been tested on human melanoma cancer A375 and HeLa cancer cell lines, which are known to overexpress BAG3 protein, by an MTT cell viability assay, using different concentrations (5–50 μM). The IC_50_ values have been calculated after 48 h of treatment with test compounds (Table 2) [26].

Molecules **5**, **6** and **8**, **9** showed promising antiproliferative effects, with IC_50_ values in the micromolar range in both cancer cell lines; among these, compound **6** displayed the best biological profile in accordance with SPR assay results (A375 **1**: IC_50_ = 15.08 ± 0.9 µM; **6**: IC_50_ = 19.36 ± 1.2 µM; HeLa **1**: IC_50_ = > 50 µM; **6**: IC_50_ = 18.67 ± 0.9 µM). Compound **6** was, thus, selected for further biological investigations.

Since BAG3 possesses an antiapoptotic activity, the effects of **6** on A375 and HeLa cell cycle distribution, by flow cytometry analysis, were analyzed. The cells were incubated for 48 h with **1** and **6** at concentrations 5-10-25μM. Upon treatment with **6,** a dose-dependent accumulation in the G_2_ phase was observed in both HeLa and A375 cells (Figure 3a and Figure 4a). Concerning **1**, an accumulation in the G_2_ phase was observed in A375 cells, while no evident effect was found in HeLa cells (Figure 3a). In agreement with these results, in both cell lines compound **6** induced a significant increase of the apoptotic response in a dose-dependent manner, as depicted in Figure 3b and Figure 4b displaying the hypodiploid nuclei. A dose-dependent increase in hypodiploid nuclei was also observed for **1** in A375 cells (Figure 3b).

Furthermore, the expression of caspases, the key mediators of programmed cells death, was examined in HeLa cells after the treatment with compound **6**. The flow cytometry analysis showed a significant and dose-dependent activation of both caspase 3 and caspase 9 levels in cells following the incubation with compound **6** (Figure 5a,b) [27,28].

Finally, an expression decrease of BAG3 was observed upon treatment with compound **6** of HeLa cells. This result confirms the direct interference of **6** with BAG3 which is known to self-regulate its levels through a positive feedback mechanism (Figure 6) [29].

Taking together, all these data point out the important role of **6** in BAG3 modulation, emerging, thus, as an attractive candidate for drug development and a useful tool for further biological investigations of this multifaceted co-chaperone. Hence, by using an integrated approach exploiting computer-aided procedures, biophysical techniques, and biological evaluation, the 2,4-thiazolidindione scaffold was successfully confirmed as a promising chemical platform able to selectively modulate BAG3 activity.

## 3. Materials and Methods

### 3.1. In-Silico Screening

The starting library of 2,4-thiazolidinedione-based compounds (~2.3 × 10^4^ items) was initially subjected to a filtering step in order to select those featured esters and amide functions at the N-3 position, using the Ligand Filtering utilities implemented in Schrödinger Suite (4242 output compounds). Afterwards, the pharmacokinetic properties of these compounds were computed using QikProp software (Schrodinger, LLC, New York, NY, USA), related to: (a) absorption, distribution, metabolism, and excretion (ADME); (b) reactive functional groups types, excluding those that may cause false positives in high-throughput screening (HTS) assays, and/or decomposition.

Specifically, the following parameters were accounted: (a) molecular weight < 400 g/mol; (b) #stars: number of property or descriptor values that fall outside the 95% range of similar values for known drugs (range or recommended values: 0–5); (c) rtvFG: number of reactive functional groups, which can lead to false positive in high-throughput screening (HTS) assays and to decomposition, reactivity, or toxicity problems in vivo (range or recommended values: 0–2); (d) #donorHB: estimated number of hydrogen bonds that would be donated by the solute to water molecules in an aqueous solution. Values are averages taken over a number of configurations, so they can be non-integer (range or recommended values: 0.0–6.0); (e) #accptHB: estimated number of hydrogen bonds that would be accepted by the solute from water molecules in an aqueous solution. Values are averages taken over a number of configurations, so they can be non-integer (range or recommended values 2.0–20.0); (f) #QPlogPo/w: predicted octanol/water partition coefficient (range or recommended values −2.0–6.5).

After further “qualitative” filtering steps (i.e., compatibility with synthetic procedure, commercial availability of the synthons, etc., see Results and Discussion), a fast “shape screening” alignment was then performed, using the 3D structure of the reference BAG3 lead inhibitor **1** as shape query ligand. Phase software (Schrodinger, LLC, New York, NY, USA) was employed [19]; in particular, for the screened compounds, the sampling was performed allowing the conformers around the amide bond to vary freely, and finally, 1000 maximum number of conformers were considered for the shape computation. Once associated a shape similarity value for each compound computed against **1**, a ranking from the best to worst values was obtained, saving those featured having a shape similarity value higher than 0.500.

Further filters, specifically aimed to select a focused set of compounds useful for both assessing the predicted binding with the protein counterpart and to facilitate the definition of a minimal structure-activity relationship, were applied, leading to eight items (compounds **2**–**9**) as promising 2,4-thiazolidinedione-based compounds for the subsequent chemical synthesis and biological evaluation steps.

### 3.2. Synthesis of 2,4-Thiazolidinedione Molecules

All commercial chemicals and solvents were purchased from Merck (Darmstadt, Germany) and were used as received. Solvents used for the synthesis were of High-performance liquid chromatography (HPLC) grade (Merck). Nuclear Magnetic Resonance (NMR) spectra (^1^H, ^13^C) were recorded on a Bruker Avance 400 MHz instrument (Bruker, Milan, Italy) at T = 298 K. Compounds were dissolved in 0.5 mL of CD_3_OD, CDCl_3_, or (CD_3_)_2_SO (Merck, 99.8 Atom %D). NMR chemical shifts were reported in δ and are expressed in parts per million (ppm) relative to the solvent peak as internal reference; coupling constants (*J*) are expressed in Hertz. Multiplicities are reported as follows: s, singlet; d, doublet; dd, doublet of doublets; t, triplet; dt, doublet of triplets; q, quartet; m, multiplet. Mass spectrometry experiments were performed using a Q-ToF^TM^ (Quadrupole time of flight) Premiere instrument (Waters, Co.^®^, Milan, Italy) equipped with an ESI (Electrospray ionization) source, or using an LTQ Orbitrap XL^TM^ mass spectrometer (Thermo Scientific^TM^, Monza, Italy). Chemical reactions were monitored on silica gel 60 F254 plates (Merck) and the spots were visualized under UV (Ultraviolet radiation) light (λ = 254 nm, 365 nm). Analytical and semi-preparative reversed-phase HPLC were performed on an Agilent Technologies, Inc.© (Agilent, Cernusco sul Naviglio, MI, Italy) 1200 Series high performance liquid chromatography using a Nucleodur© C8 reversed-phase column (Phenomenex, Castel Maggiore, BO, Italy) (100 × 2 mm, 4μM, 80 Å, flow rate = 1 mL/min; 250 × 10.00 mm, 4μM, 80 Å, flow rate = 4 mL/min respectively, Macherey-Nagel©). The binary solvent system (A/B) was as follows: 0.1% TFA (Trifluoacetic acid) in water (A) and 0.1% TFA in CH_3_CN (B). The absorbance was detected at λ = 240 nm. All biologically tested compounds were determined to be >98% pure by HPLC analysis and NMR data.


*General procedure (A) for the synthesis of *
**10a**
*–*
**10d**
Thiazolidine-2,4-dione **10** (1.45 mmol; 1.0 equiv), the appropriate aromatic aldehydes **a**–**d** (1.45 mmol; 1.0 equiv), piperidine (1.17 mmol; 0.8 equiv), and ethanol (5 mL) were placed in a 25-mL round-bottom flask. The reaction mixture was continuously stirred and refluxed for 16–24 h (Figure 1) [22]. The crude product was poured into water and extracted with AcOEt (3 × 25 mL). The combined organic phases were washed with water (3 × 25.0 mL), dried over anhydrous Na_2_SO_4_, filtered, and concentrated in vacuum. The desired compounds **10a**–**10d** were confirmed by analytical RP-HPLC (Reversed-Phase-High Performance Liquid Chromatography) (Nucleodur© C8 reversed-phase column: 100 × 2 mm, 4 μM, 80 Å, flow rate = 1 mL/min) (Phenomenex, Castel Maggiore, BO, Italy) and ^1^H NMR spectra. **10a**–**10d** were used without any further purification in the next step.
*(Z)-5-(3,4-dihydroxybenzylidene)thiazolidine-2,4-dione (*
**10a**
*)*
Compound **10a** was obtained by following the general procedure (A), from the reaction between **10** and 3,4-dihydroxybenzaldehyde (**a**), as a brown solid (310 mg, 85% yield by HPLC analysis). RP-HPLC t_R_ = 21.5, gradient condition: from 5% B ending to 100% B 50 min, flow rate of 4 mL/min, λ = 240 nm. ^1^H NMR (400 MHz, CD_3_OD): δ_H_ = 7.55 (s, 1H), 6.91 (d, *J* = 2.2 Hz, 1H), 6.86 (dd, *J* = 8.3, 2.2 Hz, 1H), 6.77 (d, *J* = 8.3 Hz, 1H). ESI-MS: calculated for C_10_H_7_NO_4_S 237.01 found *m*/*z* = 238.00 [M + H]^+^.
*(Z)-5-(3,5-dihydroxybenzylidene)thiazolidine-2,4-dione (*
**10b**
*)*
Compound **10b** was obtained by following the general procedure (A), from the reaction between **10** and 3,5-dihydroxybenzaldehyde (**b**), as a brown solid (340 mg, 80% yield by HPLC analysis). RP-HPLC t_R_ = 21.5, gradient condition: from 5% B ending to 100% B 50 min, flow rate of 4 mL/min, λ = 240 nm. ^1^H NMR (400 MHz, CD_3_OD): δ_H_ = 7.51 (s, 1H), 6.39 (d, *J* = 2.1 Hz, 2H), 6.25 (t, *J* = 2.1 Hz, 1H). ESI-MS: calculated for C_10_H_7_NO_4_S 237.01 found *m*/*z* = 238.02 [M + H]^+^.
*(Z)-5-(4-(bis(2-hydroxyethyl)amino)benzylidene)thiazolidine-2,4-dione *
*(*
**10c**
*)*
Compound **10c** was prepared in the same manner described for the preparation of **10a** following the general procedure (A). From the reaction between **10** and 4-[*N,N*-Bis(2-hydroxyethyl)amino]benzaldehyde (**c**), **10c** was obtained as a brown solid (420 mg, 80% yield by HPLC analysis). RP-HPLC t_R_ = 20.4, gradient condition: from 5% B ending to 100% B 50 min, flow rate of 4 mL/min, λ = 240 nm. ^1^H NMR (400 MHz, CD_3_OD): δ_H_ = 7.53 (s, 1H), 7.30 (d, *J* = 8.9 Hz, 2H), 6.76 (dd, *J* = 7.4, 5.6 Hz, 2H), 3.66 (t, *J* = 5.7 Hz, 4H), 3.53 (t, *J* = 6.0 Hz, 4H). ESI-MS (Electrospray Ionization-Mass Spectrometry): calculated for C_14_H_16_N_2_O_4_S 308.08 found *m*/*z* = 309.13 [M + H]^+^.
*(Z)-4-(6-((2,4-dioxothiazolidin-5-ylidene)methyl)pyridin-3-yl)benzonitrile (*
**10d**
*)*
Compound **10d** was obtained by following the general procedure (A), from the reaction between **10** and 4-(6-formylpyridin-2-yl)benzonitrile **(d)**, as a brown solid (430 mg, 85% yield by HPLC analysis). RP-HPLC t_R_ = 32.5, gradient condition: from 5% B ending to 100% B 50 min, flow rate of 4 mL/min, λ = 240 nm. ^1^H NMR (400 MHz, CD_3_OD): δ_H_ = 8.43 (d, *J* = 8.1 Hz, 2H), 7.98–7.87 (m, 4H), 7.66 (s, 1H). 7.62 (d, *J* = 7.2 Hz, 1H). ESI-MS: calculated for C_16_H_9_N_3_O_2_S 307.04 found *m*/*z* = 308.11 [M + H]^+^.
*General procedure (B) for the synthesis of *
**11**
*,*
**2**
*–*
**4**
*, and*
**6**
*–*
**9**
In a two-neck round-bottom flask, sodium hydride NaH (1.2 equiv) was added portion-wise to a solution of **10a**–**10d** (0.07 g; 1.0 equiv) in dry DMF (3.5 mL). The reaction mixture was stirred for 1.5 h at 80 °C and it was allowed to return to room temperature. Then, ethyl bromoacetate (**e**) (1.5 equiv) for the compounds **3**, **6**, **8**, methyl bromoacetate (**f**) (1.5 equiv) for **2**, **4**, **7**, **9**, and bromoacetic acid (1.5 equiv) (**g**) for the synthesis of **11** was added and the mixture was stirred at 80°C for 16–20 h. The reaction was monitored by TLC (Thin-layer chromatography) (Synthetic Figure 1) [22,23]. The suspension was poured into ice-cold water and the solid product was filtered and recovered. HPLC purification was performed by semi-preparative reversed-phase HPLC (Nucleodur© C8 reversed-phase column: 250 × 10.00 mm, 4 μM, 80 Å, flow rate = 4 mL/min) and the final products were characterized by ESI-MS and NMR spectra.
*(Z)-methyl 2-(5-(3,4-dihydroxybenzylidene)-2,4-dioxothiazolidin-3-yl) acetate (*
**2**
*)*
Compound **2** was obtained by following the general procedure (B), from the reaction between **10a** and methyl bromoacetate (**f**), as a brown solid (75 mg, 50% yield after HPLC purification). RP-HPLC t_R_ = 28.9 min, gradient condition: from 5% B ending to 100% B 50 min, flow rate of 4 mL/min, λ = 240 nm. ^1^H NMR (400 MHz, CD_3_OD): δ_H_ = 7.82 (s, 1H), 7.07 (d, *J* = 2.2 Hz, 1H), 7.02 (dd, *J* = 8.4, 2.2, 1H), 6.91 (d, *J* = 8.2 Hz, 1H), 4.52 (s, 2H), 3,79 (s, 3H). ^13^C NMR (100 MHz, CD_3_OD): δ = 168.79, 168.71, 166.95, 150.05, 146.97, 136.02, 126.10, 125.51, 117.55, 117.33, 116.74, 53.00, 42.52. ESI-MS: calculated for C_13_H_11_NO_6_S, 309.03; found *m*/*z* = 332.02 [M + Na]^+^.
*(Z)-ethyl 2-(5-(3,5-dihydroxybenzylidene)-2,4-dioxothiazolidin-3-yl) acetate (*
**3**
*)*
Compound **3** was obtained by following the general procedure (B), from the reaction between **10b** and ethyl bromoacetate (**e**), as a brown solid (80 mg, 60% yield after HPLC purification). RP-HPLC t_R_ = 28.1 min, gradient condition: from 5% B ending to 100% B 50 min, flow rate of 4 mL/min, λ = 240 nm. ^1^H NMR (400 MHz, (CD_3_)_2_SO): δ_H_ = 9.72 (s, 2H), 7.80 (s, 1H), 6.52 (d, *J* = 6.9 Hz, 2H), 6.38 (d, *J* = 7.2 Hz, 1H), 4.49 (s, 2H), 4.18 (q, *J* = 7.3 Hz, 2H), 1.23 (t, *J* = 7.3 Hz, 3H). ^13^C NMR (100 MHz, CD_3_OD): δ = 167.38, 166.89, 165.54, 159.09 (2C), 134.67, 134.44, 120.59, 108.13 (2C), 105.00, 61.69, 41.60, 13.04. ESI-MS: calculated for C_14_H_13_NO_6_S, 323.05; found *m*/*z* = 346.03 [M + Na]^+^.
*(Z)-methyl 2-(5-(3,5-dihydroxybenzylidene)-2,4-dioxothiazolidin-3-yl) acetate (*
**4**
*)*
Compound **4** was obtained by following the general procedure (B), from the reaction between **10b** and methyl bromoacetate (**f**), as a brown solid (75 mg, 75% yield after HPLC purification). RP-HPLC t_R_ = 29.5 min, gradient condition: from 5% B ending to 100% B 50 min, flow rate of 4 mL/min, λ = 240 nm. ^1^H NMR (400 MHz, CD_3_OD): δ_H_ = 7.78 (s, 1H), 6.54 (d, *J* = 2.1 Hz, 2H), 6.39 (t, *J* = 2.1 Hz, 1H), 4.53 (s, 2H), 3.80 (s, 3H). ^13^C NMR (100 MHz, CD_3_OD): δ = 167.42, 167.38, 165.50, 159.07 (2C), 134.66, 134.45, 120.52, 108.24 (2C), 104.95, 51.85, 41.37. ESI-MS: calculated for C_13_H_11_NO_6_S, 309.03; found *m*/*z* = 332.02 [M + Na]^+^.
*(Z)-ethyl 2-(5-(4-(bis(2-hydroxyethyl) amino) benzylidene)-2,4-dioxothiazolidin-3-yl) acetate (*
**6**
*)*
Compound **6** was obtained by following the general procedure (B), from the reaction between **10c** and ethyl bromoacetate (**e**), as a yellow solid (60 mg, 45% yield after HPLC purification). RP-HPLC t_R_ = 26.2 min, gradient condition: from 5% B ending to 100% B 50 min, flow rate of 4 mL/min, λ = 240 nm. ^1^H NMR (400 MHz, CD_3_OD): δ_H_ = 7.84 (s, 1H), 7.49–7.45 (m, 2H), 6.93–6.90 (m, 2H), 4.49 (s, 2H), 4.28–4.22 (m, 2H), 3.79 (t, *J* = 6.0, 4H), 3.67 (t, *J* = 6.0 Hz, 4H), 1.33–1.28 (m, 3H). ^13^C NMR (100 MHz, CD_3_OD): δ = 167.8, 167.1, 166.0, 150.4, 134.9, 132.5 (2C), 120.3, 113.0, 111.8 (2C), 61.6, 58.8 (2C), 53.3 (2C), 41.4, 12.88. ESI-MS: calculated for C_18_H_22_N_2_O_6_S, 394.12; found *m*/*z* = 395.23 [M + H]^+^.
*(Z)-methyl 2-(5-(4-(bis(2-hydroxyethyl) amino) benzylidene)-2,4-dioxothiazolidin-3-yl) acetate (*
**7**
*)*
Compound **7** was obtained by following the general procedure (B), from the reaction between **10c** and methyl bromoacetate (**f**), as a yellow solid (70 mg, 40% yield after HPLC purification). RP-HPLC t_R_ = 24.2 min, gradient condition: from 5% B ending to 100% B 50 min, flow rate of 4 mL/min, λ = 240 nm. ^1^H NMR (400 MHz, CD_3_OD): δ_H_ = 7.84 (s, 1H), 7.47 (d, *J* = 8.6 Hz, 2H), 6.91 (d, *J* = 8.6 Hz, 2H), 4.51 (s, 2H), 3.81–3.77 (m, 7H), 3.67 (t, *J* = 5.7 Hz, 4H), 1.31 (s, 2H). ^13^C NMR (100 MHz, CD_3_OD): δ = 169.0, 168.8, 167.1, 151.6, 136.2, 133.7 (2C), 121.6, 114.2, 113.1 (2C), 60.0 (2C), 54.5 (2C), 53.0, 42.5. ESI-MS: calculated for C_17_H_20_N_2_O_6_S, 380.10; found *m*/*z* = 381.20 [M + H]^+^.
*(Z)-ethyl 2-(5-((5-(4-cyanophenyl) pyridin-2-yl) methylene)-2,4-dioxothiazolidin-3-yl) acetate (*
**8**
*)*
Compound **8** was obtained by following the general procedure (B), from the reaction between **10d** and ethyl bromoacetate (**e**), as a yellow solid (100 mg, 50% yield after HPLC purification). RP-HPLC t_R_ = 42.3 min, gradient condition: from 5% B ending to 100% B 50 min, flow rate of 4 mL/min, λ = 240 nm. ^1^H NMR (400 MHz, CDCl_3_): δ_H_ = 8.24 (d, *J* = 8.3 Hz, 2H), 7.97–7.92 (m, 2H), 7.87 (d, *J* = 8.2 Hz, 2H), 7.80 (d, *J* = 8.0 Hz, 1H), 7.58 (d, *J* = 7.6 Hz, 1H), 4.52 (s, 2H), 4.27 (q, *J* = 7.1 Hz, 2H), 1.32 (t, *J* = 7.1 Hz, 3H). ^13^C NMR (100 MHz, CDCl_3_): δ = 170.8, 166.3, 165.8, 155.9, 151.8, 142.3, 138.3, 132.8 (2C), 129.3, 127.8 (2C), 126.9, 126.6, 121.1, 118.6, 113.3, 62.2, 41.7, 14.06. ESI-MS: calculated for C_20_H_15_N_3_O_4_S, 393.08; found *m*/*z* = 416.03 [M + Na]^+^.
*(Z)-methyl 2-(5-((5-(4-cyanophenyl) pyridin-2-yl) methylene)-2,4-dioxothiazolidin-3-yl) acetate (*
**9**
*)*
Compound **9** was obtained by following the general procedure (B), from the reaction between **10d** and methyl bromoacetate (**f**), as a dark brown solid (75 mg, 45% yield after HPLC purification). RP-HPLC t_R_ = 43.2 min, gradient condition: from 5% B ending to 100% B 50 min, flow rate of 4 mL/min, λ = 240 nm. ^1^H NMR (400 MHz, CDCl_3_): δ_H_ = 8.17–8.14 (m, 2H), 7.89–7.84 (m, 2H), 7.80–7.76 (m, 2H), 7.73 (dd, *J* = 8.0, 0.9 Hz, 1H), 7.50 (dd, *J* = 7.7, 0.9 Hz, 1H), 4.45 (s, 2H), 3.73 (s, 3H). ^13^C NMR (100 MHz, (CD_3_)_2_SO): δ = 170.5, 167.7, 165.5, 155.2, 151.5, 142.2, 139.5, 133.3 (2C), 130.4, 128.5, 128.1 (2C), 125.3, 122.6, 119.1, 112.6, 53.1, 42.0. ESI-MS: calculated for C_19_H_13_N_3_O_4_S, 379.06; found *m*/*z* = 380.21 [M + H]^+^.
*(Z)-2-(5-(3,5-dihydroxybenzylidene)-2,4-dioxothiazolidin-3-yl) acetic acid (*
**11**
*)*
Compound **11** was obtained by following the general procedure (B), from the reaction between **10a** and bromoacetic acid (**g**), as a brown solid (90 mg, 70% yield after HPLC purification). RP-HPLC t_R_ = 16.3 gradient condition: from 5% B ending to 100% B 50 min, flow rate of 4 mL/min, λ = 240 nm. ^1^H NMR (400 MHz, CD_3_OD): δ_H_ = 7.79 (s, 1H), 6.62 (d, *J* = 2.0 Hz, 2H), 6.26 (t, *J* = 1.9 Hz, 1H), 4.63 (s, 2H). ESI-MS: calculated for C_12_H_9_NO_6_S, 295.02; found *m*/*z* = 296.00 [M + Na]^+^. The desired compound was directly used for the following step.
*(Z)-5-(3,5-dihydroxybenzylidene)-3-(2-morpholino-2-oxoethyl) thiazolidine-2,4-dione (*
**5**
*)*
Compounds 1-Hydroxybenzotriazole (0.48 mmol; 2.0 equiv) and *N*,*N*′-diisopropylcarbodiimide (0.36 mmol; 1.5 equiv) were added to a solution of **11** (0.24 mmol; 1.0 equiv) in DMF (3 mL). The mixture was stirred for 1h at room temperature and then morpholine (0.48 mmol; 2.0 equiv) was added. The reaction mixture was stirred overnight at room temperature to obtain **5** (Figure 1) [30]. After completion, the suspension was diluted with 10 mL of water and extracted with AcOEt (3 × 20 mL). The combined organic layers were dried over Na_2_SO_4_ and evaporated under vacuum. A portion of the crude product was purified by semi-preparative reversed-phase HPLC (Nucleodur© C8 reversed-phase column: 250 × 10.00 mm, 4 μM, 80 Å, flow rate = 4 mL/min). The desired compound **5** was characterized by ESI-MS and NMR spectra.Compound **5** was obtained as a pale brown solid (85 mg, 55% yield after HPLC purification). RP-HPLC t_R_ = 17.5 min, gradient condition: from 5% B ending to 100% B 50 min, flow rate of 4 mL/min, λ = 240 nm. ^1^H NMR (400 MHz, CD_3_OD): δ_H_ = 7.71 (s, 1H), 6.57 (s, 2H), 6.40–6.37 (m, 2H), 6.26 (dt, *J* = 6.1, 2.2 Hz, 1H), 5.34 (s, 2H), 3.61–3.54 (m, 4H), 3.52–3.45 (m, 4H). ^13^C NMR (100 MHz, CD_3_OD): δ = 171.9, 170.7, 169.3, 159.0 (2C), 132.9, 132.2, 120.5, 108.2 (2C), 104.9, 75.9 (2C), 66.2 (2C), 29.4. ESI-MS: calculated for C_16_H_16_N_2_O_6_S, 364.07; found *m*/*z* = 362.90 [M − H]^+^.

### 3.3. Biophysical Assay

The SPR analysis were performed on Biacore 3000^TM^ equipped with a research-grade CM5 sensor chip (GE Healthcare^TM^, Milan, Italy). Human recombinant BAG3 (rBAG3) protein was purchased from MyBioSource© (San Diego, CA, USA), BAG3 domain (rBAG3BD) was purchased from ARETA International S.r.l. (Gerenzano, VA, Italy), and human recombinant BAG4 protein was purchased from Abnova, Co© (Taipei, Taiwan) Our previously synthesized lead compound **1** was used as a positive control [17]. The proteins were coupled to the surface of a CM5 sensor chip using standard amine-coupling protocols, according to the manufacturer’s instructions. Three separates recombinant rBAG proteins surfaces and one unmodified reference surface were prepared for simultaneous analyses. rBAG proteins (BAG3 and BAG4) and rBAG3BD (100 µg mL^−1^ in 10 mM CH_3_COONa, pH 4.5) were immobilized on the sensor chip surface at a flow rate of 5 µL min^−1^ to obtain densities of 5–9 kRU. Compounds **1**–**9** were dissolved in 100% DMSO to obtain 50 mM solutions and diluted 1:100 (*v*/*v*) in PBS (phosphate-buffered saline: 10 mM NaH_2_PO_4_, 150 mM NaCl, pH 7.4) to a final DMSO (dimethyl sulfoxide) concentration of 1.0%. For each molecule, a six-point concentration series was conducted in triplicate (0-0.25-1-10-20-100 µM). SPR (Surface Plasmon Resonance) experiments were performed at 25 °C, using a flow rate of 10 µL min^−1^, with 60-s monitoring of association and 300-s monitoring of dissociation. Sensorgram elaborations were performed using the BIAevaluation software (GE Healthcare^TM^, Milan, Italy). provided by GE Healthcare^TM^. Simple interactions were suitably fitted to a single-site bimolecular interaction model (A + B = AB), yielding a single K_D_.

### 3.4. Biological Assays

#### 3.4.1. Cell Culture

Cell lines of human cervical carcinoma (HeLa) and human malignant melanoma (A375) were cultured in Dulbecco’s modified Eagle’s medium containing 10% fetal bovine serum, supplemented with 100 U/mL each of penicillin and streptomycin, and 2 mM/L glutamine and were grown at 37 °C under 5% CO_2_ air humidified atmosphere.

#### 3.4.2. Cell Viability Assay

Cell viability was evaluated using a colorimetric assay based on MTT ((3-(4,5-dimethylthiazol-2-yl)-2,5-diphenyltetrazolium bromide)) assay. Cells were grown in 96-well tissue culture plates (3,5 × 10^3^ cells/well) and after 24 h, were treated with different concentrations of **1**, **2** and **5**–**9** (50, 25, 10, and 5 µM) and incubation was performed for 48 h at 37 °C in an atmosphere containing 5% CO_2_. At the end of treatment, the plates were centrifuged at 1200 rpm for 5 min, the medium was removed and 100 μL of MTT (1 mg/mL) were added to each well, and cells were incubated at 37 °C for the time necessary to allow the formation of purple formazan precipitate. The solution was then removed from each well, and the formazan crystals within the cells were dissolved with 100 μL of DMSO [31]. The optical density (OD) of each well was measured with a microplate spectrophotometer (Multiskan Spectrum Thermo Electron Corporation reader) (Thermo Fisher Scientific, Rome, Italy) equipped with a 620-nm filter. Cell mortality was calculated as: % mortality = 100 − [100 × (OD treated/OD control)] [32].

#### 3.4.3. Apoptosis and Cell Cycle Analysis

The effect of compound **1** and **6** on cell death was analyzed by propidium iodide (PI) (Merck, Darmstadt, Germany) incorporation in permeabilized cells and flow cytometry. Briefly, cells were plated at a density of 2 × 10^4^ cells/well in a 24-well plate. After 24 h, **1** and **6** (25-10-5 μM) were added for 48 h. Cells were washed twice with PBS and incubated in 500μL of a solution containing 0.1% Triton X-100, 0.1% sodium citrate, and 50 mg/mL propidium iodide (PI), at 4 °C for 30 min in the dark. The PI-stained cells were subsequently analyzed by flow cytometry by FACSCalibur flow cytometer (Becton Dickinson, Milan, Italy) using CellQuest software (4.0, Becton Dickinson, North Ryde, NSW, Australia). Results are expressed as the percentage of cells in the hypodiploid region. Cellular debris were excluded from the analysis by raising the forward scatter threshold and the DNA content of the nuclei was registered on a logarithmic scale. Cell cycle profiles were evaluated by DNA staining with a PI solution using a flow cytometer [33]. Results from 5000 events per sample were collected, and the relative percentage of the cells in G_0_/G_1_, S, G_2_/M, and sub-G_0_/G_1_ phases of the cell cycle was determined using the ModFit LT version 3.3 analysis software (BD Biosciences, North Ryde, NSW, Australia).

#### 3.4.4. Measurement of Caspase 3 and Caspase 9 Levels

Cells were plated into 12-well plated (2 × 10^4^ cells/well) and after 24 h were treated with **6** (50-25-10-5 µM) for 48 h to assess caspase 3 and caspase 9 levels. Cells were collected, washed with PBS, then incubated in fixing buffer (containing 4% formaldehyde, 0.1% NaN_3_, and 2% FBS in PBS) for 20 min and then permeabilized with fix perm solution (fixing buffer containing 0.1% Triton X) for 30 min. Afterward, anti-caspase 3 or anti-caspase 9 antibodies were added for a further 30 min. Cells were then incubated with anti-rabbit Texas Red antibody as a secondary antibody for 1 h at 4 °C. Cell fluorescence was evaluated using a fluorescence-activated cell sorter (FACSscan; Becton Dickinson, Milan, Italy) and analyzed with the Cell Quest software (4.0, Becton Dickinson, North Ryde, NSW, Australia). Data were shown as a percentage of positive cells.

#### 3.4.5. Data Analysis

Data are reported as mean ± S.E.M. values of independent experiments, performed at least three times, with three or more independent observations. Statistical analysis was performed by Student’s *t*-test. Differences with *p* < 0.05 were considered statistically significant.

## 4. Conclusions

Starting from the first disclosed BAG3 modulator **1**, previously discovered by us, a new collection of differently decorated compounds featuring a 2,4-thiazolidinedione core was synthesized. The affinity towards the target of interest was evaluated by performing SPR assays using both the full-length BAG3 protein and the isolated BAG domain. By a deeply biological evaluation of the selected molecules, compound **6** was disclosed as a new interesting antiproliferative agent able to interfere with BAG3 functions. These promising results may stimulate further research toward the identification of more potent and selective BAG3 binders as attractive candidates for the development of new anticancer drugs.

## Data Availability

All relevant data are presented within the body of this paper.

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
