# Peer review of "Structural Refinement of 2,4-Thiazolidinedione Derivatives as New Anticancer Agents Able to Modulate the BAG3 Protein"

_molecules, 2022, doi:10.3390/molecules27030665_

Round 1
Reviewer 1 Report
Review report for Manuscript ID: molecules-1544794
Title: Structural refinement of 2,4-thiazolidinedione derivatives as new anticancer agents able to modulate BAG3 protein by Ines Bruno et al. The authors have carried out in silico-selection of 2,4-thiazolidinediones & evaluated their binding affinity for the BAG3BD domain.
General comments-
Insert DOI for all citations in the references section.
Author Response
DOI for all citations in the references section have been included as required.
Reviewer 2 Report
Authors have reported the synthesis of 2,4-thiazolidinedione derivatives as new anticancer agents that are able to modulate BAG3 protein. BAG3 has emerged as a potential and effective biological target to control multiple cancer-related signalling pathways. The topic is original as they reported the first synthetic BAG3 modulator interfering with its BAG domain (BD), based on a 2,4-thiazolidinedione scaffold and endowed with significant anti- proliferative activity and performed silico studies.
Some specific improvements can be done to improve the quality of the paper
1. Authors should include the most active derivative along with IC50 value in the abstract section.
2. In the procedure line 408-409 include (mmol; equivalents) for each compound and also mention in other synthetic procedures.
3. For compds 10a -10d (carbon NMR may be given).
4. In the proton NMR data please specify the type of proton whether aromatic proton or aliphatic proton it need to be distinguish
5. Author reported the compounds to be anticancer effect on cancer line they may also either check on cytotoxic effects on Normal Cell lines or give some references for similar derivatives
Author Response
We are grateful for the suggestions that contributed to improve the quality of the manuscript.
All the requested revisions have been made and here reported point by point:
1.Rev.: Authors should include the most active derivative along with IC50 value in the abstract section.
Auth.: We added in the abstract section the most active derivative along with IC50 value (lines 23-25)
2.Rev.: In the procedure line 408-409 include (mmol; equivalents) for each compound and also mention in other synthetic procedures.
Auth.: We made the required revision
3.Rev.: For compds 10a -10d (carbon NMR may be given).
Auth.: Since compounds 10a-10d are intermediate compounds, as it is usually reported we include 1H NMR and MS data that, generally are fully accepted for their structural identification.
4.Rev.: In the proton NMR data please specify the type of proton whether aromatic proton or aliphatic proton it need to be distinguish
Auth.: We made the required revision
5.Rev.: Author reported the compounds to be anticancer effect on cancer line they may also either check on cytotoxic effects on Normal Cell lines or give some references for similar derivatives.
Auth.: Concerning this point, the reference number 18, including the data of the effect on Normal Cell lines of the lead compound 1 can be taken into account.